



**Revealing the Influence of Topography and Vegetation on**
**Hydrological Processes Using a Stepwise Modelling Approach in Cold**
**Alpine Basins of the Mongolian Plateau**
Leilei Yong [1], Yahui Wang [1], Batsuren Dorjsuren [2], Zheng Duan [3], Hongkai Gao [1]*
[1] Key Laboratory of Geographic Information Science (Ministry of Education of China),
School of Geographical Sciences, East China Normal University, Shanghai, China
[2] Department of Environment and Forest Engineering, National University of Mongolia,
Ulaanbaatar 210646, Mongolia
[3] Department of Physical Geography and Ecosystem Science, Lund University,
Sölvegatan 12, SE-223 62, Lund, Sweden
* Correspondence: Hongkai Gao, hkgao@geo.ecnu.edu.cn
**Abstract:** Topography and vegetation are critical factors influencing catchment
hydrology; however, their individual contributions are often underestimated in
hydrological models. This limitation is particularly evident in cold, mountainous
regions such as the Mongolian Plateau, where observational data are sparse. To address
this, we employed a stepwise, top-down modelling strategy based on the FLEX
framework to systematically assess the influence of topography and vegetation on
hydrological processes in the Bogd Uliastai and Zavkhan Guulin river basins.
Beginning with a lumped model (FLEX$^L$), we successively integrated snow processes
(FLEX$^L$-S), topographic distribution (FLEX$^D$), and finally, a landscape-based
parameterization accounting for vegetation heterogeneity (FLEX$^T$). Both FLEX$^D$ and
FLEX$^T$ outperformed the lumped models in simulating runoff and SWE. Interestingly,
FLEX$^T$ showed similar performance to FLEX$^D$ — likely due to limited vegetation
heterogeneity — it offers more physically realistic parameterization by explicitly
representing landscape units, suggesting its potential in more complex basins.
Snowmelt contributions to streamflow were quantified as 23.5%±1.3% and 14.7%±1.6%





in the Bogd Uliastai and Zavkhan Guulin river basins, respectively, with peaks in spring
and a clear increase with elevation. At high elevations, delayed snowmelt resulted in
sustained runoff, while lower elevations responded more rapidly to rainfall. The explicit
representation of vegetation heterogeneity further improved the model's capacity to
capture landscape complexity and dominant hydrological mechanisms. This study
underscores the pivotal roles of topography and vegetation in runoff generation and
demonstrates the effectiveness of a stepwise modelling framework for improving
hydrological understanding in cryospheric and data-scarce regions.
**Keywords:** Mongolian Plateau, FLEX model, stepwise modelling framework,
snowmelt, topography, vegetation

## 1. Introduction

Understanding and accurately simulating hydrological processes are fundamental for
elucidating basin hydrological patterns and supporting water resource management and
ecological protection, especially under the context of global environmental change
(Gomes et al., 2023; Oki and Kanae, 2006). Topography and vegetation play essential
roles as drivers of hydrological processes, influencing key aspects such as precipitation,
interception (Dwarakish and Ganasri, 2015), snowmelt (Hammond et al., 2019),
evaporation (Jiao et al., 2017), and runoff generation (Qin et al., 2025). Topography
governs water flow paths and moisture release processes (Gao et al., 2014), while
vegetation affects water movement and infiltration by regulating precipitation
interception and soil moisture dynamics (Zhu et al., 2022). The complex interaction
between topography and vegetation not only define Hydrological Response Units
(HRUs) but also shape the spatial heterogeneity and dominant hydrological
mechanisms within a river basin (Savenije, 2010; Sivapalan, 2009). However, in cold-
arid regions, data scarcity often leads to oversimplified hydrological models, limiting
accurate simulations (Ragettli et al., 2014; Tarasova et al., 2016). Therefore, a more
comprehensive evaluation of topography – vegetation interactions is essential for
improving model fidelity and supporting effective water resource management and

https://doi.org/10.5194/egusphere-2025-3062





ecological conservation.

Topography plays a fundamental role in shaping hydrological processes by influencing
the spatial distribution of soil moisture, regulating precipitation patterns, modulating
evaporation dynamics, and driving runoff generation, thereby governing the movement
and storage of water across the landscape (Wicki et al., 2023). In mountainous basins,
variations in topographic relief introduce substantial uncertainty into hydrological
modeling (Seibert and McDonnell, 2002). Steeper slopes typically lead to more rapid
runoff, while gentler slopes promote greater infiltration and moisture retention, thus
affecting the spatial and temporal distribution of water resources (Ye et al., 2023).
Moreover, topography critically influences snow distribution and snowmelt dynamics.
Terrain features such as slope, aspect, and elevation induce spatial heterogeneity in
snow accumulation and melting processes, resulting in diverse hydrological responses
across the basin (Broxton et al., 2020).

Vegetation plays a crucial role in regulating hydrological processes, particularly
through interception and root zone water storage. First, vegetation canopies intercept
rainfall, reducing the amount of effective precipitation reaching the soil, while also
mitigating surface erosion and slowing runoff (Cheng et al., 2020). Second, root zone
storage capacity and plant transpiration regulate soil moisture, enhance evaporation and
facilitating water redistribution (Luo et al., 2022; Volpe et al., 2013). These effects vary
by vegetation type, as different structural forms (e.g., forests vs. grasslands) exhibit
distinct hydrological behaviors (Chen et al., 2023). In cold mountainous regions,
vegetation also affects snow processes by affecting snow distribution and retention. For
example, forest canopies can shield snow accumulation, delay snowmelt, and reduce
wind-driven redistribution, thereby significantly altering the spatiotemporal dynamics
of meltwater runoff (Sun et al., 2022).

Although the regulatory role of topography and vegetation in basin hydrology are



widely acknowledged,, their synergistic interactions remain insufficiently understood,
particularly in cold high-altitude mountainous regions characterized by complex terrain,
harsh climatic conditions, and limited observational data (Stephens et al., 2021).
Cryospheric regions serve as critical freshwater resources for downstream areas and are
especially sensitive to changes in the hydrological cycle and climate (Immerzeel et al.,
2010). In these regions, snowfall and snowmelt processes often dominate runoff
generation, with topography and vegetation jointly modulating hydrological responses
by influencing snow distribution, accumulation, and melt rates (Dharmadasa et al.,
2023; Zhong et al., 2021). Therefore, quantifying the individual and combined effects
of topography and vegetation, and effectively integrating them into hydrological
models, is essential for advancing cold-region hydrology.

Existing hydrological models often struggle to adequately capture the complexities
introduced by topography and vegetation. Early lumped models typically used basin-
averaged precipitation and temperature to simulate runoff, thereby oversimplifying
spatial heterogeneity within catchments (Beven, 2012). While computationally efficient,
lumped models fail to accurately represent the spatial variability of terrain and land
cover, especially in mountainous regions. The advent of distributed hydrological
models has allowed more spatially explicit simulations by incorporating topographic
and land cover data (Fenicia et al., 2016). However, the performance of these models
is highly dependent on data quality, which remains a significant limitation in cold, high-
mountain regions where traditional observations are sparse or unavailable.

Remote sensing has become an invaluable tool for providing high-resolution data on
topography, vegetation, and snow in hydrological studies of cold regions. Digital
elevation models (DEMs) offer critical topographic information such as slope, aspect,
and elevation, while vegetation indices derived from remote sensing (e.g., NDVI and
EVI) effectively characterize vegetation cover (Xiong et al., 2023). In addition, remote
sensing techniques enable spatial monitoring of snow water equivalent and snowmelt





processes (Duethmann et al., 2014). Integrating remote sensing data with distributed
hydrological models helps to overcome the limitations of traditional in-situ
observations, offering a more comprehensive understanding of the roles that
topography and vegetation play in shaping hydrological processes (Gao et al., 2014).

In the absence of direct measurements of individual hydrological processes, the top-
down modelling approach offers a powerful means of exploring the internal dynamics
of basin behavior (Fenicia et al., 2008b). Originally proposed by Klemes (Klemeš, 1983)
and later reformulated by Sivapalan et al. (Sivapalan et al., 2003), the top-down
approach is rooted in a deductive philosophy that infers the underlying 'causes' from
the overall observed 'effect' of a system. In hydrological modeling, this method begins
with a simple structure that is progressively refined to address limitations in
reproducing observed catchment behavior (Fenicia et al., 2008a). In cold mountainous
regions, the top-down approach holds significant potential for improving model realism
by systematically incorporating key variables such as snow processes, topography, and
vegetation.

This study focuses on the Bogd Uliastai and Zavkhan Guulin river basins on the
Mongolian Plateau, aiming to investigate the roles of topography and vegetation in
shaping hydrological processes in cold mountainous regions. Due to the scarcity of
observational data, traditional hydrological models face significant challenges in these
areas. To address this, we employ a top-down modelling approach, beginning with a
lumped model to assess runoff dynamics and progressively advancing toward a
distributed framework. This model explicitly incorporates key components—including
snowmelt, topography, and vegetation—to better capture the hydrological responses of
different landscape units. The study seeks to address three key research questions: (1)
How can runoff be effectively simulated in data-scarce, cold mountainous regions using
a top-down modelling approach? (2) How can the contribution of snowmelt to
streamflow be quantified through a landscape-based hydrological model? (3) How do



topography and vegetation influence runoff generation processes?

## 2. Study site

### 2.1 Bogd Uliastai river basin

The Bogd Uliastai river basin (47°30′N-48°10′N, 96°45′E-97°45′E) is located in the
northern part of the Zavkhan river headwaters, along the southern foothills of the central
Khangaï Mountains in Mongolia (Fig.1). The basin spans an area of 1610 km$^2$ and is
predominantly mountainous, with elevations ranging from 1753 m a.s.l to 3972 m a.s.l.
The region receives an average annual precipitation of approximately 200 mm, with
more than 80% of rainfall occurring between June and September. The average annual
temperature is -1°C, while winter temperatures frequently fall below -30°C, reflecting
a typical alpine climate. Runoff displays strong seasonal variability, with distinct peaks
during the spring and summer and almost no flow in winter, resulting in extreme
hydrological conditions (Dorjsuren et al., 2024). The vegetation exhibits clear
altitudinal zonation: alpine meadows and tundra, dominated by mosses and lichens,
prevail at higher elevations, whereas needlegrass steppe and low shrublands are
common in mid- and low-elevation zones (Baasanmunkh et al., 2019).





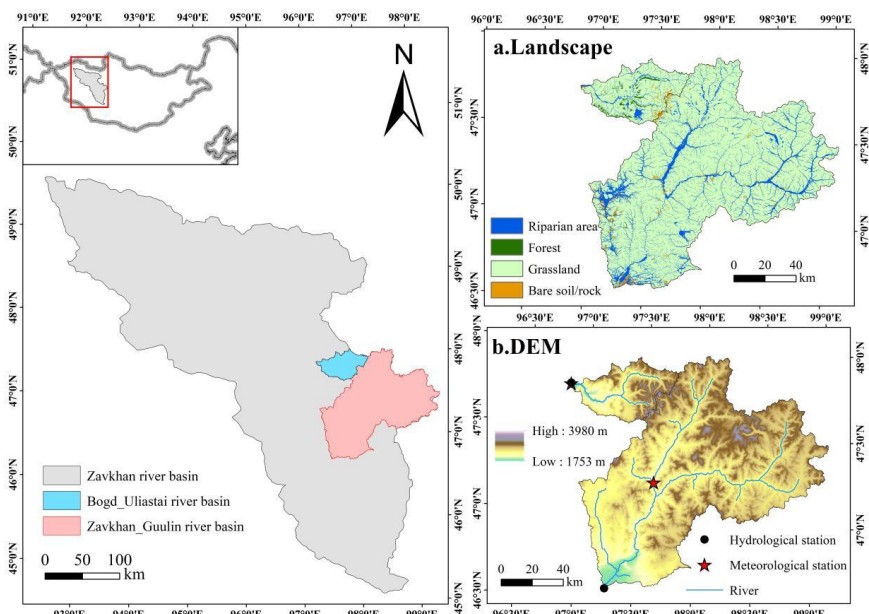

**Fig.1** Location, landscape (a) and topography (b) of the Bogd Uliastai and Zavkhan Guulin river
basins on the Mongolian Plateau.

## 2.2 Zavkhan Guulin river basin

The Zavkhan Guulin river basin (46°30′N-47°50′N, 96°45′E-97°00′E), located in the

central and southern parts of Zavkhan Province, lies within the transitional zone of the

southern Khangaï Mountains (Fig.1). The basin covers an area of approximately 12258

km² and is predominantly composed of low mountains and hills, with elevations

ranging from 1785 m a.s.l. to 3980 m a.s.l. The basin's annual average precipitation is

about 160 mm, with most precipitation concentrated in the summer, primaril in the form

of heavy rain, which serves the main source of runoff. The annual average temperature

is approximately -3°C, with summer temperatures exceeding 20°C and winter

temperatures dropping as low as -50°C, characteristic of a temperate continental climate

(Dorjsuren et al., 2023). Vegetation in the region is sparse, primarily dominated by

drought-tolerant *Artemisia* species, with scattered distributions of grass and shrubs. At

higher elevations, the landscape is characterized by alpine meadows, exposed rock

surfaces, and cold desert environments. Soils are nutrient-poor, and the ecological



environment is fragile, facing severe challenges such as soil erosion (Baasanmunkh et
al., 2019).

**3. Data**
**3.1 Data set**
**Hydrometeorological data:** Daily precipitation, runoff, and temperature data for the
Bogd Uliastai river basin (2007–2015) and the Zavkhan Guulin river basin (2000–2020)
were obtained from the Information and Research Institute of Meteorology, Hydrology,
and Environment (IRIMHE) via its official website (http://irimhe.namem.gov.mn). For
each basin, one meteorological station and one hydrological station served as the
primary sources of observational data. The Arctic Snow Water Equivalent (SWE) Grid
Dataset (2003–2016) was obtained from National Tibetan Plateau/Third Pole
Environment Data Center (https://cstr.cn/18406.11.Snow.tpdc.271556). The SWE
product has a daily temporal resolution and a spatial resolution of 10 km, covering
latitudes from 45°N to 90°N and longitudes from 180°W to 180°E.
**Topographic data:** The Shuttle Radar Topography Mission Digital Elevation Model
(SRTM-DEM), with a spatial resolution of 30 m, was acquired from the official website
of the International Center for Tropical Agriculture (http://srtm.csi.cgiar.org).
**Land cover data:** The Sentinel-2 10-Meter Land Use/Land Cover was accessed via
ESRI's official platform (https://livingatlas.arcgis.com/landcover/).
**NDVI data:** The normalized difference vegetation index (NDVI) data (2013–2020)
were derived from the Landsat 8 Operational Land Imager (OLI) Level-2 surface
reflectance products. NDVI was calculated as (NIR – Red)/(NIR + Red) using bands
5 (NIR) and 4 (Red). The dataset has a spatial resolution of 30 m and a temporal
resolution of 16 days. Landsat data were obtained from the United States Geological
Survey (USGS) EarthExplorer platform (https://earthexplorer.usgs.gov/).
**3.2 Distribution of forcing data**





Mountainous terrain is complex, and meteorological stations are typically located at
lower elevations. Directly using point-based measurement in basin-scale simulations
without accounting for elevation effects can introduce biases (Klemeš, 1989). In cold
mountainous regions, higher elevations typically experience lower temperatures and
greater precipitation, often in the form of snow (Lundquist et al., 2010; Stahl et al.,
2006). In the study, the FLEX model divides catchment into elevation bands and adjusts
temperature and precipitation for each band using a precipitation increase rate and
temperature lapse rate. This distributed input approach effectively mitigates simulation
bias by better capturing altitudinal variability in meteorological conditions. In this study,
due to the remoteness of the region and the sparse distribution of meteorological
stations, available ground observations were limited. Satellite and reanalysis products
(e.g., ERA5) exhibit notable biases over complex terrain and fail to capture local
climatic variability. We therefore relied on the best available in-situ data, which were
subjected to rigorous quality control and spatial interpolation, and supplemented by
topographic context and previous studies. While uncertainties remain, this approach
provides the most reliable climate forcing achievable under current observational
constraints. The model employed a precipitation increase rate of 4.2% per 100 m and a
temperature lapse rate of 0.6°C per 100 m (Gao et al., 2014).

## 4. Modelling approach

### 4.1 Model description

To assess the impact of topography and vegetation on hydrological processes, this study
designed and tested four conceptual models with increasing complexity: $FLEX^L$,
$FLEX^L$-S, $FLEX^D$, and $FLEX^T$ (Fig.2). The model structure and variables are shown in
Fig. 2 and Table 1, and the water balance, isotope mass balance and constitutive
equations are shown in Table 2.



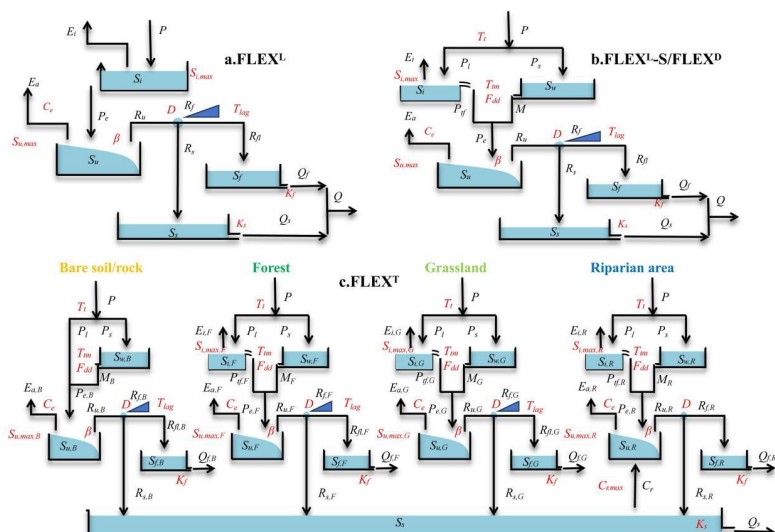


**Fig.2** Stepwise modelling and the model structure of four models. (a) FLEX$^L$ is a lumped model without snow module; (b) FLEX$^L$-S is a lumped model with snow module, and FLEX$^D$ is a semi-distributed model with the same structure as FLEX$^L$-S. (c) FLEX$^T$ is a landscape-driven semi-distributed model.

**Table1.** The variables of four models. In FLEX$^T$ model, variables associated with various landscape categories are differentiated using specific suffixes, e.g., $E_{i,F}$, represent the interception from forest.

| Variables | Meaning | Variables | Meaning |
|---|---|---|---|
| $P$ (mm/d) | Precipitation | $E_i$ (mm/d) | Interception |
| $S_i$ (mm) | Interception reservoir | $P_s$ (mm/d) | Snowfall |
| $P_l$ (mm/d) | Rainfall | $P_{tf}$ (mm/d) | Effective rainfall after interception |
| $M$ (mm/d) | Snowmelt | $P_e$ (mm/d) | Effective precipitation |
| $S_u$ (mm) | Unsaturated reservoir | $E_a$ (mm/d) | Actual evaporation |
| $R_u$ (mm/d) | Generated runoff from the unsaturated reservoir | $R_f$ (mm/d) | Generated fast runoff in the unsaturated zone |
| $R_{fl}$ (mm/d) | Discharge into the fast response reservoir after the convolution | $R_s$ (mm/d) | Generated slow runoff in the unsaturated zone |
| $S_f$ (mm) | Fast response reservoir | $S_s$ (mm) | Slow response reservoir |
| $C_r$ (mm/d) | Capillary rise from groundwater into unsaturated reservoir on riparian area | $Q_f$ (mm/d) | Subsurface storm flow |
| $Q_s$ (mm/d) | Groundwater flow | $Q$ (mm/d) | Total runoff |




**Table 2.** The water balance and constitutive equations used in four models. Note: FLEX$^L$ model lacks the snow module, resulting in different water balance and structural equations compared to other models. For FLEX$^T$ model, in Eqs.(4), (6), and (7), the $S_i$ and $S_{i,max}$ represent interception reservoir and their interception capacities in different landscapes, including forest ($S_{i,max,F}$), grassland ($S_{i,max,G}$) and riparian area ($S_{i,max,R}$) (There is no interception store in bare soil/rock area.). Similarly, in Eqs. (8), (10), (11), and (12), the $S_u$ and $S_{u,max}$ represent root zone reservoirs and their storage capacities in different landscapes, including bare soil/rock ($S_{u,max,B}$), forest ($S_{u,max,F}$), grassland ($S_{u,max,G}$) and riparian area ($S_{u,max,R}$).

| Reservoirs | Water balance equations | Constitutive equations |
|---|---|---|
| Interception reservoir<br><br>(FLEX$^L$) | $\dfrac{dS_i}{dt} = P - E_i - P_e$ (1) | $E_i = min\left(E_p, min(P, S_{i,max})\right)$ (2)<br><br>$P_e = max(P - E_i, 0)$ (3) |
| Snow reservoir<br><br>(FLEX$^L$-S/FLEX$^D$/FLEX$^T$) | $\dfrac{dS_w}{dt} = P_s - M$ (4) | $P_s = \begin{cases} P; & T < T_t \\ 0; & T \geq T_t \end{cases}$ (5)<br><br>$M = \begin{cases} F_{dd}(T - T_{tm}); & T > T_{tm} \\ 0; & T \leq T_{tm} \end{cases}$ (6)<br><br>$P_l = \begin{cases} P; & T \geq T_t \\ 0; & T < T_t \end{cases}$ (8) |
| | $\dfrac{dS_i}{dt} = P_l - E_i - P_{tf}$ (7) | |
| Interception reservoir<br><br>(FLEX$^L$-S/FLEX$^D$/FLEX$^T$) | | $E_i = min\left(E_p, min(P_l, S_{i,max})\right)$ (9)<br><br>$P_{tf} = max(P_l - E_i, 0)$ (10)<br><br>$P_e = P_{tf} + M$ (11) |




**Unsaturated root zone reservoir (All)**

$$\frac{dS_u}{dt} = P_e - E_a - R_u \quad (12)$$

$$E_a = (E_p - E_i)\min\left(\frac{S_u}{C_e S_{u,max}}, 1\right) \quad (13)$$

$$R_u = \begin{cases} P_e - S_{u,max} + S_u + S_{u,max}\left(1 - \dfrac{P_e+AU}{(1+\beta)S_{u,max}}\right)^{(1+\beta)} & ; \; (1+\beta)S_{u,max} > P_e + AU \\[2mm] P_e - S_{u,max} + S_u; & (1+\beta)S_{u,max} \le P_e + AU \end{cases} \quad (14)$$

$$AU = (1+\beta)S_{u,max}\left(1 - \left(1 - \frac{S_u}{S_{u,max}}\right)^{\frac{1}{1+\beta}}\right) \quad (15)$$

**Splitter and lag function (All)**

$$R_f = R_u D \quad (16)$$

$$R_s = R_u(1-D) \quad (17)$$

$$R_{fl} = \sum_{i=1}^{Tlag} c(i) \cdot R_f(t-i+1) \quad (18)$$

$$c(i) = i/\sum_{u=1}^{Tlag} u \quad (19)$$

**Fast reservoir (All)**

$$\frac{dS_f}{dt} = R_f - Q_f \quad (20)$$

$$Q_f = S_f/K_f \quad (21)$$

**Slow reservoir (All)**

$$\frac{dS_s}{dt} = R_s - Q_s \quad (22)$$

$$Q_s = S_s/K_s \quad (23)$$





### 4.1.1 FLEX$^L$

FLEX$^L$ is a lumped conceptual hydrological model composed of four reservoirs (Fig.2a): an interception reservoir ($S_i$), an unsaturated reservoir ($S_u$), a fast response reservoir ($S_f$), and a slow response reservoir ($S_s$). A lag function is used to represent the lag time from storm to peak flow ($T_{lag}$). FLEX$^L$ includes a total of 8 free calibration parameters (Table 3).

The interception reservoir was designed to simulate the process of precipitation interception by vegetation canopies or the ground surface (Eq.1). Interception evaporation ($E_i$) was calculated by potential evaporation ($E_p$) and $S_i$, considering the interception storage capacity ($S_{i,max}$) (Eq.2). When precipitation ($P$) exceeds $S_{i,max}$, the excess precipitation is routed as effective precipitation ($P_e$) into the unsaturated reservoir (Eq.3).

In the unsaturated reservoir, actual evaporation ($E_a$) was estimated based on $E_p$ and root zone soil moisture ($S_u/S_{u,max}$) (Eq.13). The parameter $C_e$ represents the threshold value controlling evaporation from the root zone soil moisture, and $S_{u,max}$ is root zone storage capacity. The water retention curve from the Xin'anjiang model was used to partition $P_e$ into stored water in $S_u$ and runoff generated from the unsaturated root zone ($R_u$) (Zhao, 1992) (Eqs.14 and 15).

In the response reservoir, a splitter $D$ was applied to divide the $R_u$ into two fluxes ($R_f$ and $R_s$) (Eqs.16 and 17), and Eqs (18) and (19) were used to describe the lag time between storm and peak flow. $R_f(t\text{-}i\text{+}1)$ represents the fast runoff generated in the unsaturated zone at time $t\text{-}i\text{+}1$, $T_{lag}$ represents the time lag between the storm and fast runoff generation. $c(i)$ is the weight of the flow in $i\text{-}1$ days before and $R_{fl}(t)$ is the discharge into the fast response reservoir after convolution. We used two linear reservoirs to represent the response process of subsurface storm flow ($Q_f$) and groundwater flow ($Q_s$) (Eqs.21 and 23).





### 4.1.2 FLEX$^L$-S

FLEX$^L$-S builds upon the FLEX$^L$ model by incorporating a snow reservoir ($S_w$) to simulate the snow accumulation and melt processes (Fig.2b). When the daily air temperature exceeds the threshold temperature ($T_t$) and there is no snowpack (typically in summer), the interception process governs the initial partitioning of precipitation (Eq.7). In contrast, when the daily mean temperature is below $T_t$ (normally occurs in winter), precipitation is stored as snow (Eq.5). When there is snowpack and the daily air temperature is above $T_t$ (normally prevailing in early spring and early autumn), effective precipitation ($P_e$) is equal to the sum of effective rainfall after interception ($P_{tf}$) and snowmelt ($M$) (Eq.11). $M$ was calculated by the snow degree day factor ($F_{dd}$) and the threshold temperature for melting ($T_{tm}$) (Eq.6) (Gao et al., 2017). In this study, $T_{tm}$ was set to the same value as $T_t$. It is important to note that meltwater is conceptualized as directly infiltrating into the soil, thereby bypassing the interception reservoir.

### 4.1.3 FLEX$^D$

FLEX$^D$ is a semi-distributed model with the same structure and parameters as FLEX$^L$-S (Fig.2b). Using DEM data, the Bogd Uliastai river basin was divided into 45 elevation bands with 50 m interval, while the Zavkhan Guulin river basin was divided into 44 elevation bands as shown in Fig.3. The FLEX$^D$ model was operated with semi-distributed input data (see Sect.3.2), ensuring the integration of spatial variability into the model's processes.

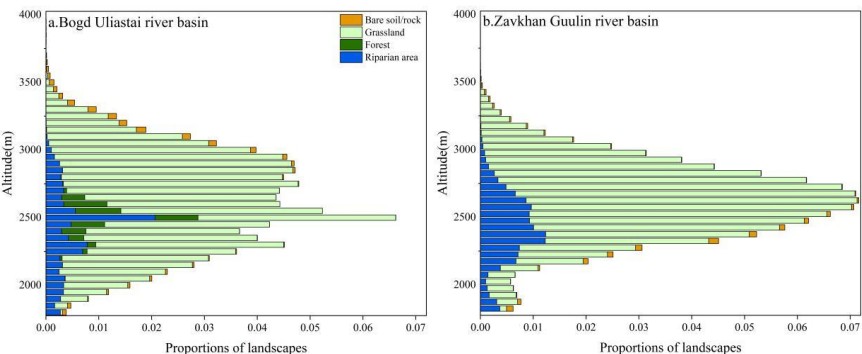



**Fig.3** Area of different elevation and landscape in Bogd Uliastai and Zavkhan Guulin river basins.

### 4.1.4 FLEX$^T$

The FLEX$^T$ model classified the Bogd Uliastai river basin into four landscape elements—bare soil/rock, forest, grassland, and riparian area—based on vegetation characteristics. In contrast, the Zavkhan Guulin river basin, which has no forest, was categorized into three landscape elements. By integrating both landscape types and elevation bands, the Bogd Uliastai river basin was further subdivided into 132 HRUs, while the Zavkhan Guulin river basin consisted of 117 HRUs (Fig.3).

The FLEX$^T$ model's structure comprised four parallel components, representing the distinct hydrological functions associated with landscape elements (Savenije, 2010; Gao et al., 2014) (Fig.2c). To capture the diverse rainfall-runoff processes in different landscape types and simultaneously avoid over-parameterization, we kept the same model structure but gave different interception storage capacity ($S_{i,max}$) and root zone storage capacity ($S_{u,max}$) for all landscape elements (Table 3).

For forest, due to their dense vegetation cover and the greater amount of water required to fill the root zone to meet water deficits, larger prior ranges were assigned to $S_{i,maxF}$ and $S_{u,maxF}$. For bare soil/rock, due to no vegetation cover, we constrained a shallower $S_{u,maxB}$ and did not incorporate an interception module. For the riparian area, which is prone to saturation due to its location, we also constrained a shallower $S_{u,maxR}$, with the effect of capillary rise ($C_r$) taken into account. $C_r$ is represented by a parameter ($C_{r,max}$) indicating a constant amount of capillary rise. Notably, the lag time from storm to peak flow was not considered in riparian area. For grasslands, $S_{u,maxG}$ is lower than that of forest but higher than bare soil/rock and riparian area.




321

**Table 3.** Uniform prior parameter distributions of four models. Note: $S_{i,max}$ and $S_{u,max}$ do not belong to the FLEX$^T$ model.

| Models | Parameter | Explanation | Prior range |
|---|---|---|---|
| FLEX$^L$ | $S_{i,max}$ (mm) | Storage capacity of interception reservoir | (0.1, 2) |
| | $S_{u,max}$ (mm) | Root zone storage capacity | (5, 300) |
| | $C_e$ (-) | Threshold controls actual evaporation and transpiration | (0, 1) |
| | $\beta$ (-) | Shape parameter of the tension water storage capacity curve | (0.1, 5) |
| | $D$ (-) | Splitter between surface runoff and groundwater recharge | (0, 1) |
| FLEX$^L$-S | $T_{lag}$ (-) | Time lag between storm and fast runoff generation | (0.8, 3) |
| FLEX$^D$ | $K_f$ (d) | Recession coefficient of fast response reservoir | (1, 10) |
| | $K_s$ (d) | Recession coefficient of slow response reservoir | (10, 200) |
| | $T_t$ (°C) | Threshold temperature to split snowfall and rainfall | (-2, 2) |
| | $T_{tm}$ (°C) | Threshold temperature for melting | (-2, 2) |
| FLEX$^T$ | $F_{dd}$ (mm(°Cd)$^{-1}$) | Snow degree day factor | (1, 6) |
| | $S_{i,max}$ (mm) | Interception storage capacity of forest | (0.5, 2) |
| | $S_{i,max}$ (mm) | Interception storage capacity of grassland | (0.1, 1) |
| | $S_{i,max}$ (mm) | Interception storage capacity of riparian area | (0.1, 1) |
| | $S_{u,maxB}$ (mm) | Root zone storage capacity of bare soil/rock | (5, 120) |
| | $S_{u,maxF}$ (mm) | Root zone storage capacity of forest | (50, 300) |
| | $S_{u,maxG}$ (mm) | Root zone storage capacity of grassland | (10, 300) |
| | $S_{u,maxR}$ (mm) | Root zone storage capacity of riparian area | (5, 120) |
| | $C_{Rmax}$ (mm/d) | Constant amount of capillary rise | (0.1, 2) |

322



**4.2 Snow contribution to streamflow**

This study tracks the contribution of snowmelt to streamflow based on FLEX$^T$. The model assumes that snowmelt and rainfall mix rapidly and completely upon entering the model's conceptual reservoirs, thereby altering its internal composition ratios. The composition ratio of the water exiting the reservoir is identical to that within the reservoir. The contributions from snowmelt and rainfall represent portions of runoff generated at each time step, with some water remaining in the reservoir to participate in subsequent mixing, runoff generation, evaporation, and other hydrological processes (Liu et al., 2023). The method enables the tracking of the contribution of snowmelt to total runoff ($C$) at each time step by the following equation:

$$C = \frac{Q_M}{Q} = \frac{Q_{f,M} + Q_{s,M}}{Q} \tag{24}$$

$$Q_{f,M} = \frac{\left(\frac{M}{P_{tf} + M}\right) S_f}{K_f} \tag{25}$$

$$Q_{s,M} = \frac{\left(\frac{M}{P_{tf} + M}\right) S_s}{K_s} \tag{26}$$

where $Q$ is total runoff in the river channel; $Q_M$ is snowmelt runoff in the river channel; $Q_{f,M}$ is subsurface storm flow generated by snowmelt; $Q_{s,M}$ is groundwater flow generated by snowmelt.

**4.3 Model calibration and uncertainty estimation**

In the Bogd Uliastai river basin, the model was pre-warmed using data from 2007; the years 2008–2011 were used for calibration, and 2012–2015 for validation. In the Zavkhan Guulin river basin, 2000 was used as the warm-up year, with 2001–2010 selected for calibration and 2011–2020 for validation.

The MOSCEM-UA (Multi-objective Shuffled Complex Evolution Metropolis Algorithm) integrates multi-objective optimization and Bayesian uncertainty analysis, featuring global search capabilities that facilitate the generation of multiple Pareto



optimal solutions and provide an assessment of uncertainty (Vrugt et al., 2003). The
MOSCEM_UA was run for the optimization of parameters, with 40000 iterations for
four model structures. The model parameters and their prior ranges for calibration are
listed in Table 3.

The Kling-Gupta Efficiency (KGE) and its logarithmic form (KGL) were used as
objective functions to evaluate the simulation of daily discharge (Gupta et al., 2009).
These two metrics were chosen because each emphasizes a different portion of the
hydrograph: KGE is more sensitive to high-flow dynamics, while KGL better captures
low-flow conditions. In this study, to accommodate minimization-based optimization
algorithms, the runoff objective functions $L_1$ (Eq.27) and $L_2$ (Eq.28) were formulated
as one minus their respective efficiency metrics. The two objective functions were
assigned equal weights during model calibration to ensure a balanced representation of
both high- and low-flow regimes.
$$L_1 = 1 - KGE = \sqrt{(1-\gamma)^2 + (1-\alpha)^2 + (1-\beta)^2} \tag{27}$$
$$L_2 = 1 - KGL = \sqrt{\left(1-\gamma_{log}\right)^2 + \left(1-\alpha_{log}\right)^2 + \left(1-\beta_{log}\right)^2} \tag{28}$$
where, $\gamma$ is the correlation coefficient between simulated and observed flows, and $\gamma_{log}$ is
the correlation coefficient between their logarithmic values; $\alpha$ is the ratio of the standard
deviations of the simulated and observed flows, and $\alpha_{log}$ is the ratio of the standard
deviations of their logarithmic transformations; $\beta$ is the ratio of the mean values of the
simulated and observed flows, and $\beta_{log}$ is the ratio of the mean values of their
logarithmic transformations.
**5. Results and discussion**
**5.1 Model calibration and validation**
Fig.4 shows the performance of the four models during the calibration period. The
Pareto-optimal front shifts progressively toward the origin, indicating that model
structural modifications enhance the model's ability to capture basin runoff dynamics.
The FLEX$^L$-S model (KGE: 0.65 and 0.65, with the former representing the Bogd
Uliastai river basin and the latter representing the Zavkhan Guulin river basin,
hereinafter referred to as the same; KGL: 0.68 and 0.66) (Table 4) outperforms the
baseline FLEX[L] model (KGE: 0.53 and 0.52; KGL: 0.62 and 0.48). This improvement
highlights the importance of explicitly representing snow processes in cryospheric
regions. Without accounting for snow accumulation and ablation, the model tends to
overestimate minor peak flow events in winter, as shown in Fig.5, underscoring the
critical role of snow dynamics in shaping hydrological responses.

The FLEX[D] model (KGE: 0.77 and 0.68; KGL: 0.74 and 0.74) outperforms the FLEX[L]-
S, with the distributed precipitation and temperature inputs significantly improving the
simulation of peak flow. Notably, FLEX[D] does not require a more complex model
structure or additional parameters compared to FLEX[L]-S. However, it allows each
hydrological response unit to maintain distinct storage states in the interception, snow,
and unsaturated reservoirs on any given day. This capability effectively overcomes a
key limitation of lumped models, which are unable to represent the spatial variability
of hydrological responses across heterogeneous landscapes.
For hydrograph simulation, FLEX[T] (KGE: 0.77 and 0.67; KGL: 0.74 and 0.75)
performs comparably to FLEX[D]. This similarity in performance—despite FLEX[T]'s
increased model complexity and more physically interpretable parameters—may be
attributed to two main factors. First, both basins are dominated by grasslands, which
cover more than 80% of the area, resulting in low vegetation heterogeneity (Fig.3).
Second, vegetation characteristics—such as rooting depth and interception capacity—
may already be implicitly represented by hydroclimatic and topographic variables
(Antonelli et al., 2018; Roebroek et al., 2020), thereby diminishing the added value of
explicitly incorporating vegetation information in this case.

As shown in Fig.6, the lumped model employs a spatially uniform NDVI value, which
cannot reflect intra-basin vegetation variability. Nevertheless, a strong correspondence





is observed between elevation and NDVI, particularly in grassland-dominated regions.
NDVI values across elevation bands closely match those of the corresponding grassland
zones, suggesting that vegetation distribution is strongly aligned with topographic
gradients. Although NDVI differs significantly between forested and bare land areas,
these land cover types occupy only a small fraction of the basin and contribute
negligibly to runoff generation. In this context, elevation can serve as a reliable proxy
for vegetation structure, effectively embedding vegetation-related hydrological
influence within the topographic representation. These findings support the notion that
hydroclimatic and terrain-based variables may indirectly encode essential vegetation
processes in distributed or semi-distributed models.

Together, these results suggest that the limited vegetation heterogeneity in the study
basins may constrain the potential performance gains of FLEX$^T$ over the simpler
FLEX$^D$ model. Nonetheless, the strength of FLEX$^T$ lies in its explicit representation of
distinct landscape units, enabling a more physically grounded simulation of
hydrological processes and underlying mechanisms. Further research is warranted to
evaluate the benefits of the landscape-based modeling approach in catchments with
greater ecological and topographic complexity.

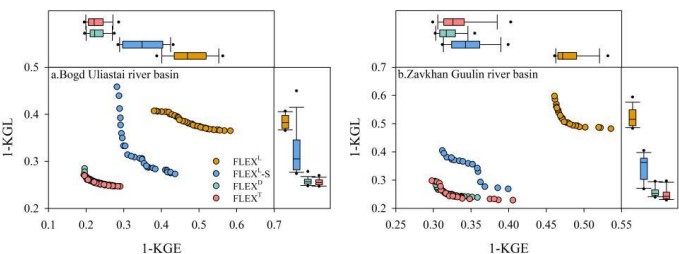


**Fig.4** Performance of the FLEX$^L$, FLEX$^L$-S, FLEX$^D$, and FLEX$^T$ models in calibration mode.





**Table 4** Comparison of simulation performance among different hydrological models in the study catchments.

| Basins | Evaluation indicators | | Calibration | | | | Validation | | | |
|---|---|---|---|---|---|---|---|---|---|---|
| | | | FLEX$^L$ | FLEX$^L$-S | FLEX$^D$ | FLEX$^T$ | FLEX$^L$ | FLEX$^L$-S | FLEX$^D$ | FLEX$^T$ |
| Bogd Uliastai river basin | KGE | Max | 0.62 | 0.72 | 0.80 | 0.81 | 0.58 | 0.63 | 0.62 | 0.63 |
| | | Median | 0.53 | 0.65 | 0.77 | 0.77 | 0.39 | 0.47 | 0.56 | 0.57 |
| | | Min | 0.42 | 0.56 | 0.73 | 0.71 | 0.16 | 0.29 | 0.52 | 0.52 |
| | KGL | Max | 0.63 | 0.73 | 0.75 | 0.75 | 0.67 | 0.76 | 0.80 | 0.80 |
| | | Median | 0.62 | 0.68 | 0.74 | 0.74 | 0.64 | 0.71 | 0.78 | 0.78 |
| | | Min | 0.59 | 0.54 | 0.72 | 0.73 | 0.61 | 0.61 | 0.75 | 0.75 |
| Zavkhan Guulin river basin | KGE | Max | 0.54 | 0.69 | 0.70 | 0.70 | 0.41 | 0.59 | 0.62 | 0.64 |
| | | Median | 0.52 | 0.65 | 0.68 | 0.67 | 0.31 | 0.47 | 0.56 | 0.56 |
| | | Min | 0.46 | 0.60 | 0.64 | 0.59 | 0.13 | 0.28 | 0.41 | 0.40 |
| | KGL | Max | 0.52 | 0.73 | 0.76 | 0.77 | 0.60 | 0.73 | 0.74 | 0.76 |
| | | Median | 0.48 | 0.66 | 0.74 | 0.75 | 0.56 | 0.66 | 0.72 | 0.73 |
| | | Min | 0.40 | 0.60 | 0.70 | 0.70 | 0.47 | 0.61 | 0.70 | 0.69 |







Some interesting rain/snowmelt-runoff events also suggest that distributed models
(FLEX$^D$ and FLEX$^T$) better capture basin hydrological processes. Two such events in
the Bogd Uliastai river basin in April 2009 and September 2011 provide compelling
evidence (Fig.5). In April 2009, despite minimal precipitation, temperatures exceeded
the melting threshold, producing only a relatively insignificant peak flow. In September
2011, despite a higher daily precipitation of 12.7 mm, no runoff peak was observed
within the basin. Lumped models failed to reproduce these dynamics accurately, instead
simulating much larger peak flows. This limitation arises because lumped models do
not account for elevation-dependent variations in temperature and precipitation type.
When the average daily temperature exceeds the rain-snow separation (snowmelt)
threshold, lumped models treat all precipitation as rain (snowmelt is assumed to occur
uniformly across the entire basin). However, snowfall may still occur at higher
elevations, where temperatures are below the threshold, resulting in limited snowmelt.
Similarly, rainfall (and corresponding snowmelt) may occur in lower elevations even
when the basin-average temperature falls below the threshold.

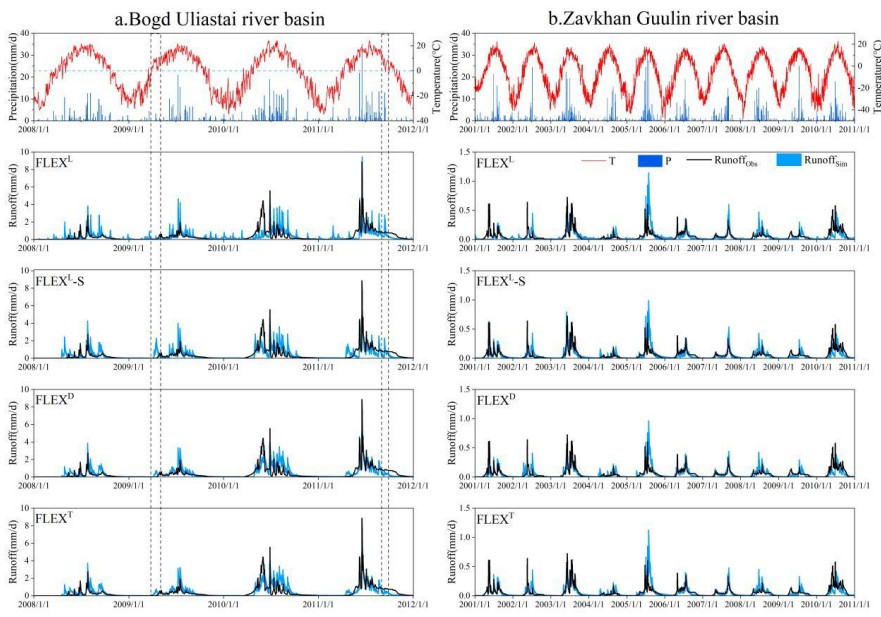


**Fig.5** The daily observed and simulated hydrographs of the FLEX$^L$, FLEX$^L$-S, FLEX$^D$, and FLEX$^T$
models in the calibration period. The dashed boxes represent the rainfall/snowfall-runoff events in



April 2009 and September 2011 in the Bogd Uliastai river basin.

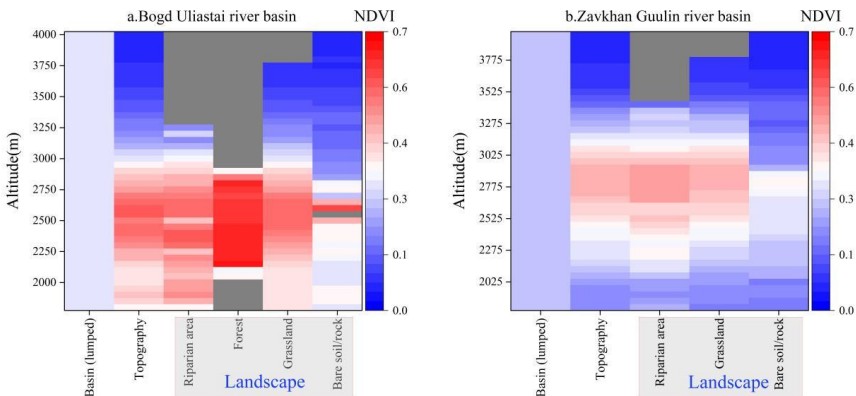


**Fig.6** Multi-year average NDVI variation across landscapes and its relationship with elevation in
two study basins.

The performance and results of the four models during the validation period are shown
in Figs.7 and 8. The results confirm the stepwise improvement in model performance,
as evidenced by the points corresponding to different model structures progressively
shifting toward the origin. With the gradual optimization of model structure, the
model's fitness has significantly improved. Unlike during calibration, the points in the
validation period do not maintain the arc shape (Fig.4). This discrepancy is attributed
to errors present in both the model and the data, the estimation of which remains a
challenging task (Fenicia et al., 2008b).

In summary, a model's ability to reproduce basin-scale hydrological responses is
governed not by the complexity of its structure or the sheer number of parameters, but
by the relevance and accuracy of the hydrological processes it represents, and their
influence on catchment-scale dynamics.





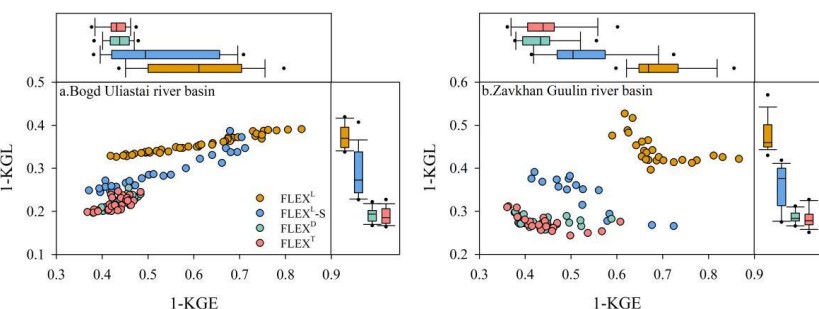

**Fig.7** Performance of the FLEX$^L$, FLEX$^L$-S, FLEX$^D$, and FLEX$^T$ models in validation mode.

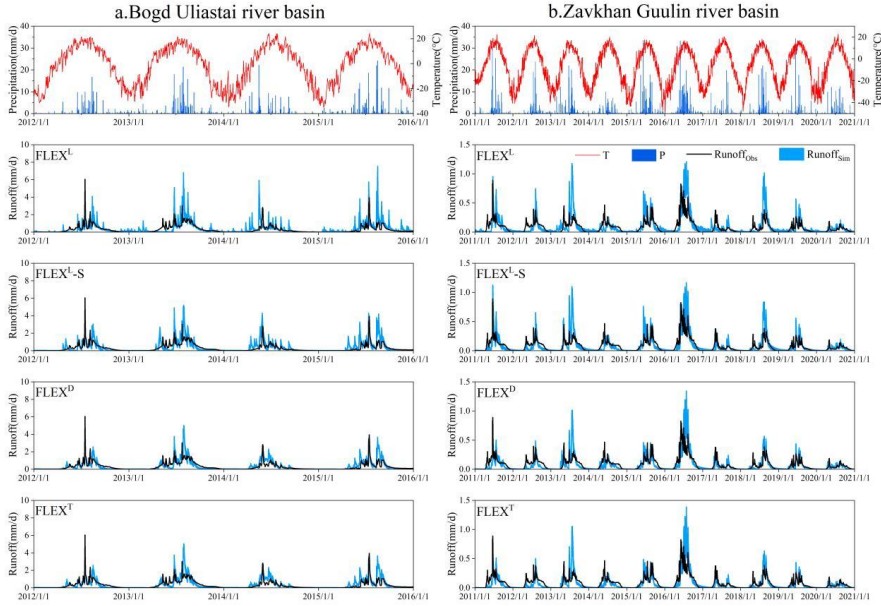

**Fig.8** The daily observed and simulated hydrographs of the FLEX$^L$, FLEX$^L$-S, FLEX$^D$, and FLEX$^T$

models in the validation period.

## 5.2 Model test by snowpack dynamics

Snow water equivalent is a crucial indicator of snowmelt dynamics and plays an

essential role in hydrological modeling, serving as an additional metric for evaluating

model performance and realism (Fig.9). In the Bogd Uliastai river basin, the FLEX$^D$

and FLEX$^T$ models achieved KGE values of 0.61 and 0.63, respectively, for SWE

simulation, indicating their ability to capture seasonal patterns and interannual



variability, particularly peak values during winter and spring. The FLEX$^T$ model, which
incorporates vegetation effects, further improved SWE simulation accuracy and
enhanced responsiveness to hydrological processes. In contrast, the FLEX$^L$-S model
yielded a KGE of only 0.37, reflecting its limitations in capturing snowpack dynamics
within the basin. Lumped models typically simplify the spatial heterogeneity of factors
such as terrain and vegetation, limiting their ability to capture local-scale features and
consequently reducing accuracy in complex environments (Bormann et al., 2009).

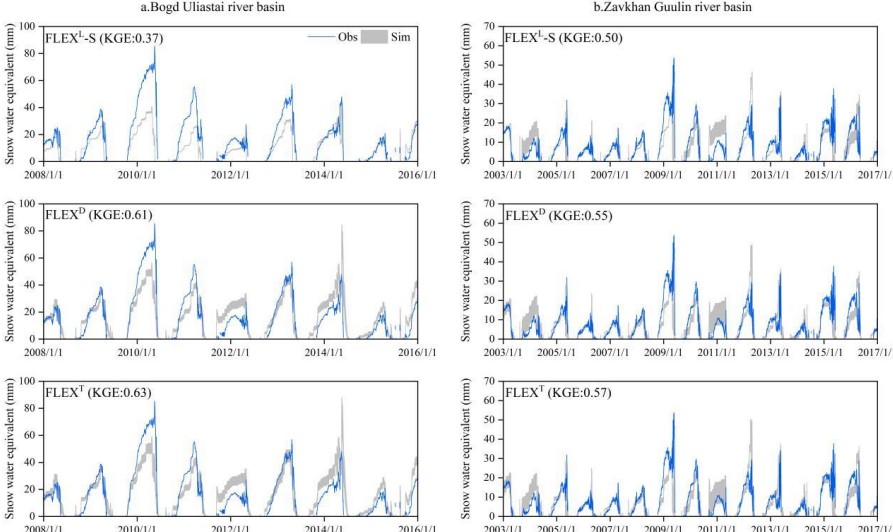


**Fig.9** The observed and simulated daily snow water equivalent of the FLEX$^L$-S, FLEX$^D$, and FLEX$^T$
models.

In the Zavkhan Guulin river basin, the FLEX$^L$-S model demonstrated relatively stable
performance, achieving a KGE of 0.50. Although lumped models struggle to capture
spatial heterogeneity, they effectively reflect seasonal precipitation and snowmelt
trends. FLEX$^D$ and FLEX$^T$ achieved KGE values of 0.55 and 0.57, respectively, showed
slight improvements. Model effectiveness remains strongly influenced by basin-
specific climatic and landscape features—such as steep slopes, variable precipitation
patterns, heterogeneous vegetation, and local climate fluctuations — all of which
complicate accurate simulation of local-scale hydrological responses (Greco et al., 2023;





Nippgen et al., 2011). These challenges are further amplified in data-scarce, cold
regions, where disentangling the interactions among these factors is particularly
difficult (Chen et al., 2017). While current models provide valuable insights, further
refinement and validation are necessary to better capture dynamic local processes and
microclimatic effects.

**5.3 Model parameters composition**
A key feature of the stepwise modelling framework is the progressive refinement of
parameterization towards greater physical realism. As shown in Fig.10, model
parameters exhibit distinct sensitivity across different structural configurations. In
models that do not account for vegetation effects, single parameter values are used to
approximate basin-average hydrological behavior. By contrast, the FLEX$^T$ model
incorporates landscape-specific hydrological response characteristics, resulting in
spatially differentiated parameter values that better reflect underlying process
heterogeneity.

Forest, characterized by dense canopies, exhibit a higher value of $S_{i,maxF}$ (1.22 mm),
effectively regulating water distribution during the initial stages of rainfall events
(Wang et al., 2021). In comparison, $S_{i,maxG}$ (0.07 mm and 0.03 mm for the Bogd Uliastai
and Zavkhan Guulin river basins, respectively) and $S_{i,maxR}$ (0.57 mm and 0.47 mm) are
lower. The riparian area, however, shows greater interception capacity than grassland,
likely due to denser or more abundant vegetation cover (Gao et al., 2014). Bare soil/rock
surfaces lack interception capacity altogether, with rainfall either infiltrating directly
into the ground or rapidly generating surface runoff along slopes.

For root zone storage capacity, $S_{u,maxF}$ is 150 mm, consistent with the findings of Wang-
Erlandsson et al. (Wang-Erlandsson et al., 2016). Notable differences are observed in
$S_{u,maxG}$ (54 mm and 283 mm) and $S_{u,maxR}$ (29 mm and 105 mm) under varying climatic
conditions. $S_{u,maxB}$ (14 mm and 33 mm) exhibits the lowest values due to the absence of



vegetation cover and limited soil structure development (He et al., 2024).
The differences in interception and root zone storage capacity across landscapes
between the two basins are primarily attributed to the more arid conditions in the
Zavkhan Guulin river basin. This basin is characterized by sparse vegetation (reflected
in lower $S_{i,max}$), higher evaporation losses (as suggested by greater $S_{u,max}$), and a low
runoff coefficient of only 0.15.

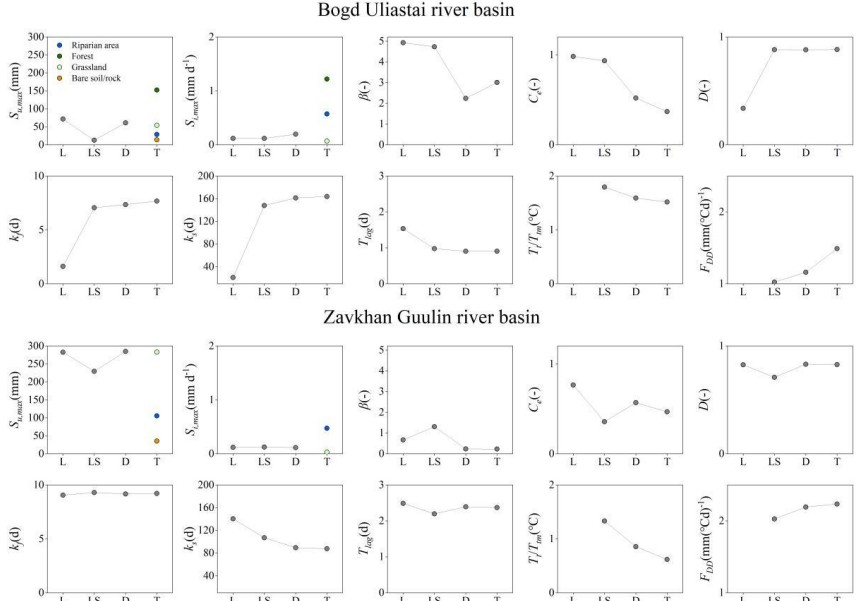


**Fig.10** The changes of averaged behavioral parameters of the FLEX$^L$, FLEX$^L$-S, FLEX$^D$, and
FLEX$^T$ models.
Other parameters are also refined alongside improvements in the model structure. The
parameter $D$, which partitions generated runoff between fast and slow response
reservoirs, tends to be close to 1—indicating that most runoff is routed to the fast
reservoir. This aligns with the observed runoff generation mechanisms in the study
basins, which are primarily driven by intense rainfall events. Parameters related to
energy processes, such as the $T_{tm}$ and $F_{dd}$, exhibit a clear compensatory relationship: a
higher $T_{tm}$ is typically associated with a lower $F_{dd}$, and vice versa. This reflects model
calibration trade-offs aimed at maintaining energy balance. Future work should



incorporate field observations to better quantify parameter heterogeneity across
different landscape units. Such efforts would enhance both the physical interpretability
and predictive robustness of the model.

### 5.4 Snow contribution to streamflow

Fig.11 shows the annual and seasonal contributions of snowmelt to streamflow in the
Bogd Uliastai and Zavkhan Guulin river basins, as determined by the FLEX$^T$ model.
On an annual scale, snowmelt contributes 23.5%±1.3% and 14.7%±1.6% to streamflow
in the Bogd Uliastai and Zavkhan Guulin river basins, respectively. Seasonally,
snowmelt plays a dominant role in sustaining spring flows, while its contribution is
considerably lower in other seasons. Although direct observational data (e.g., stable
water isotopes) for quantifying snowmelt contributions are unavailable in this study,
previous research provides indirect support. For example, Wu et al. (Wu et al., 2021)
applied a similar snowmelt tracking method in the Altai Mountain and reported that
snowmelt accounted for 29.3% of annual streamflow. This result exceeds those
observed in our study, largely due to regional differences in snowfall. In the Kayiertesi
river basin of the Altai Mountain, annual average precipitation for one hydrological
year (September to August) was 409.8 mm from 2011 to 2015 (observed at the Kuwei
snow station), with snowfall from November to March comprising about 31% of that
annual precipitation (Zhang et al., 2017). In contrast, annual precipitation in the Bogd
Uliastai and Zavkhan Guulin basins does not exceed 200 mm, and snowfall represents
less than 15% of the total observed precipitation.

This study also compared model-based snowmelt tracking with traditional indirect
methods, which estimate snowmelt contributions by calculating the ratio of snowfall or
snowmelt to runoff over a given period (Barnett et al., 2005; Immerzeel et al., 2010).
While computationally simple and data-efficient, these methods assume that all
meltwater directly contributes to runoff, neglecting interactions with rainfall and losses
due to infiltration, evaporation, and subsurface storage. Using the traditional indirect
approach, we calculated the snowmelt-to-runoff ratios to be 38.8%±2.1% and




144.4%±20.1% in the two basins, respectively. These estimates are significantly higher
than those obtained via model-based snowmelt tracking, with some values even
exceeding 100%, indicating physical implausibility. This discrepancy highlights the
limitations and scientific inadequacies of traditional methods. The overestimation likely
arises from the failure to account for spatially disconnected snowmelt—specifically,
snowmelt that infiltrates into the root zone and is subsequently lost through evaporation
(Liu et al., 2023), particularly in the arid Zavkhan Guulin river basin. These findings
underscore the importance of using physically based models to trace water source
pathways, particularly in data-scarce and hydrologically complex regions.

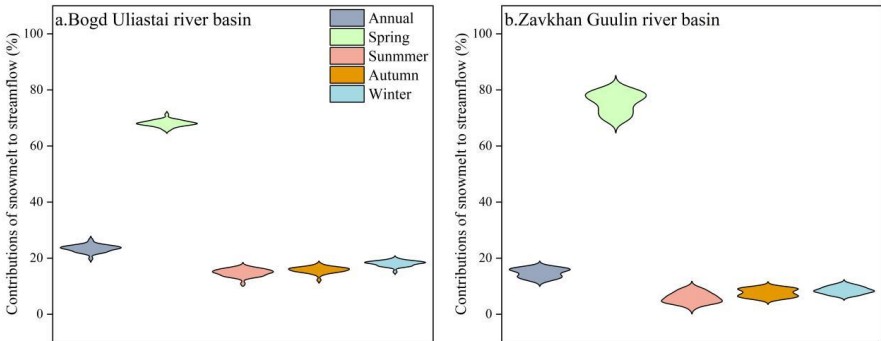


**Fig.11** Contributions of snowmelt to streamflow ($Q_M/Q$) based on the FLEX$^T$ model.

Fig.12 shows the snowmelt contribution to streamflow and the snowfall/precipitation
ratio ($P_s/P$) across different elevations. The results indicate that the $P_s/P$ increases
significantly attributable to lower temperatures at higher altitudes that favor snowfall.
Correspondingly, the contribution of snowmelt to streamflow also increases with
elevation, directly linked to greater snow storage in high elevation areas, which
providing a sustained water source for rivers (Sprenger et al., 2024). The finding
underscores the decisive influence of snowmelt on streamflow in mountainous regions.
With rising temperatures driven by climate change, low elevation areas may see more
precipitation as rain, reducing snowpack, while accelerated snowmelt at higher
elevations could increase the variability and instability of meltwater runoff



(Kraaijenbrink et al., 2021; Li et al., 2017).

Although the two basins share similar elevation and temperature regimes, their
contrasting hydrological responses primarily reflect differences in climate and
vegetation cover. The Bogd Uliastai river basin, dominated by mountainous grasslands,
exhibits higher vegetation density (basin-average NDVI: 0.31; grassland NDVI: 0.34),
whereas the Zavkhan Guulin river basin, situated in a semi-arid region, shows lower
vegetation cover (basin-average NDVI: 0.26; grassland NDVI: 0.28) (Fig.6).

These vegetation differences influence snowmelt-runoff efficiency. In Bogd Uliastai
river basin, the snowmelt contribution to streamflow closely matches the snowfall-to-
precipitation ratio, indicating limited losses and effective runoff generation. By contrast,
Zavkhan Guulin river basin experiences greater hydrological losses—primarily due to
infiltration and evaporation—which cause snowmelt contributions to fall below the
snowfall input, especially at higher elevations (Litaor et al., 2008).

Sparser vegetation and drier soils in Zavkhan Guulin river basin further enhance soil
moisture retention, delaying runoff initiation and reducing the proportion of meltwater
reaching the stream. This comparison highlights how subtle variations in vegetation
structure, captured by NDVI, modulate hydrological partitioning and runoff efficiency
across cold alpine landscapes (Zhong et al., 2021).



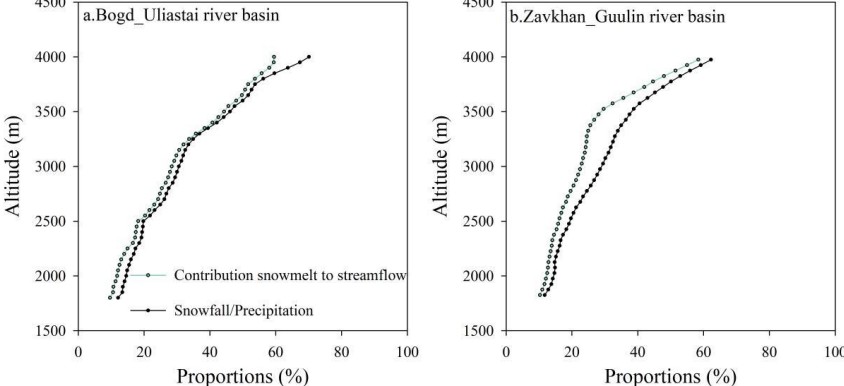

**Fig.12** Contributions of snowmelt to streamflow ($Q_M/Q$) and snowfall/precipitation ratio ($P_s/P$) at different elevations based on FLEX$^T$ model.

## 5.5 Runoff generation mechanisms at different elevation zones

Elevation is a key topographic factor influencing basin runoff processes and their seasonal variability, as it affects precipitation patterns, snow storage, and melt rates (Jenicek and Ledvinka, 2020). Fig.13 shows significant differences in runoff contributions across 5 equal area elevation bands. High elevation areas (above 2900 m or 2825 m) play a dominant role in runoff generation, primarily due to the orographic effect, which leads to increased precipitation and a higher proportion of snowmelt contributions (Ayala et al., 2023) (Fig.12). Runoff peaks in these high elevation areas are especially pronounced in spring and summer, highlighting the critical role of seasonal snowmelt. In contrast, low elevation areas rely primarily on rainfall-induced runoff. Due to limited precipitation and higher evaporative losses, their contributions to total runoff are comparatively smaller (Sprenger et al., 2022).

The lag effect in runoff is a notable characteristic of hydrological processes in mountainous basins, reflecting the differential responses of various elevation areas to hydrological drivers. In the Bogd Uliastai river basin, low elevation areas respond rapidly to precipitation events, contributing significantly to runoff during the early stages of peak flow. As the event progresses, contributions from higher elevation areas



gradually increase, highlighting the heterogeneous influence of elevation on runoff
dynamics (Hajika et al., 2024). This lag is closely associated with delayed snowmelt
process in high elevation areas. where lower temperatures cause precipitation to fall as
snow. Snowmelt in these areas typically occurs weeks or even months later than in
lower elevations, with the delay especially evident during the initial stages of the melt
season (Gillan et al., 2010). A similar pattern is observed in the Zavkhan Guulin river
basin, where runoff from high elevation continues to contribute significantly during the
latter part of the hydrograph, thereby prolonging the recession phase (Fig.13).

The lag effect of runoff across different elevation areas has important implications for
water resource management. In cold, high-mountain basins, the delayed hydrological
response of upper elevation not only sustains downstream water supply during dry
periods, but also significantly influences the timing and spatial extent of flood risk (Gu
et al., 2023). During extreme precipitation events, rapid runoff generation in low-
elevation areas may exacerbate short-term flood hazards, while delayed snowmelt from
higher elevations can prolong flood durations. Therefore, both immediate and delayed
hydrological responses should be holistically considered in catchment-scale
management strategies (Li et al., 2019)

Although the lag effect is particularly evident during runoff peaks, current observational
and modeling data remain insufficient to accurately quantify the specific response
timings and processes across elevation gradients. Future research should integrate high-
resolution numerical simulations with field-based observations to better disentangle the
dynamic runoff contributions from different elevation areas, thereby enhancing the
predictive skill and physical realism of hydrological models.





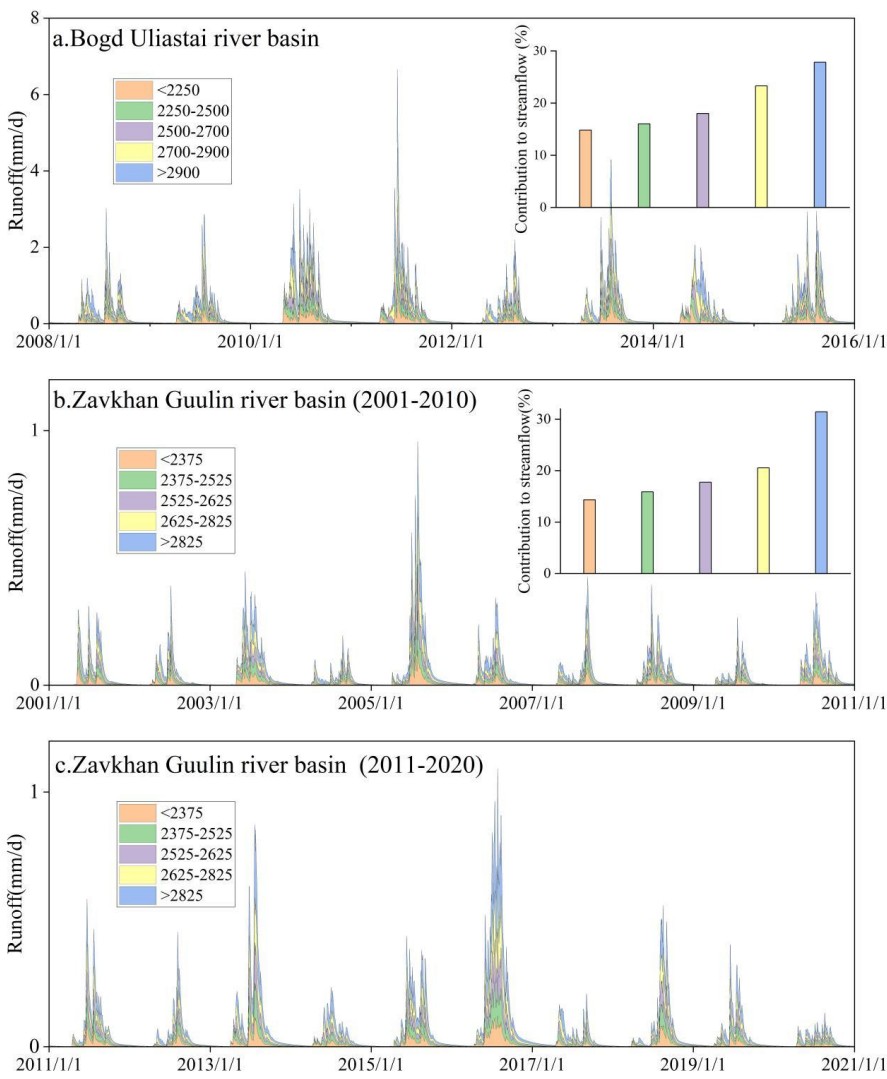

**Fig.13** Runoff contribution from 5 equal area elevation bands (each representing 20% of the total catchment area) based on FLEX$^T$ model.

## 5.6 Regulatory mechanisms of vegetation in runoff generation processes

Vegetation influences runoff generation not only through its direct physiological properties (e.g., interception, root zone storage, and transpiration), but also through its



spatial interactions with topography. To examine this regulatory role, we analyzed
runoff responses across HRUs delineated by distinct vegetation types and elevation
bands, thereby isolating process-driven variability from effects primarily driven by
areal extent (Fig.14).
Despite grasslands occupying the largest portion of both basins (82.8% and 85.4%),
their runoff generation capacity varies markedly with elevation and climatic context. In
the more arid Zavkhan Guulin river basin, dry grassland soils exhibit high infiltration
rates under unsaturated conditions, thereby reducing surface runoff. However, during
peak melt or rainfall events, saturation thresholds are exceeded, triggering rapid surface
runoff (Assouline et al., 2024).

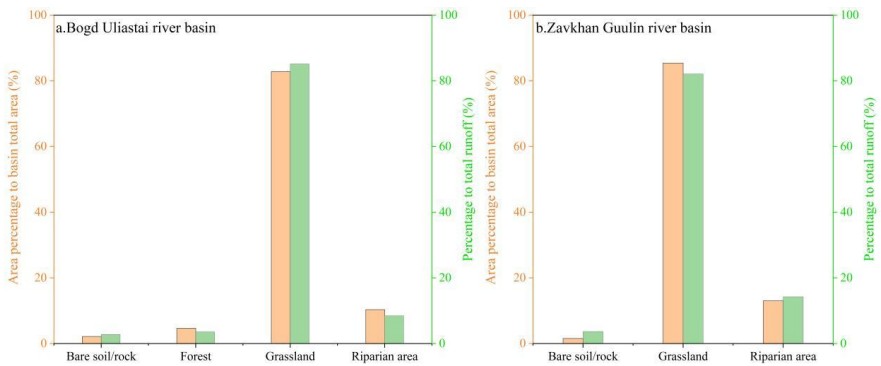

**Fig.14** Runoff contribution from different landscapes based on FLEX$^T$ model.
Riparian zones, although limited in area (10.3% and 13.0%), contribute
disproportionately to runoff (8.5% and 14.2%) due to high soil moisture, shallow root
zones, and strong hydrological connectivity. This is consistent with findings that
riparian areas function as dynamic runoff buffers, responding rapidly to precipitation
and snowmelt inputs (Leibowitz et al., 2023).
Forested areas, found only in the Bogd Uliastai basin, exhibit strong regulatory
functions—intercepting precipitation, enhancing infiltration, and reducing quick flow
generation (Liu et al., 2018; Stocker et al., 2023). These effects are particularly relevant
under scenarios of climate-induced vegetation change.




The runoff generation capacity of bare soi/rock is high due to the lack of vegetation and
low soil permeability. After rainfall or snowmelt, water infiltrates poorly and rapidly
forms overland flow (Zeng et al., 2024). However, bare surfaces cover only a small
portion of the catchment (2.2% and 1.6%), so their overall contribution to streamflow
remains limited (2.8% and 3.7%). Despite their distinct hydrological behavior, bare
areas play a secondary but notable role in runoff dynamics.

Importantly, our simulations reveal that interactions between vegetation and
topography play a critical role in shaping runoff dynamics. At higher elevations, where
vegetation is sparse and terrain is steep, snowmelt is rapidly converted into surface
runoff due to limited soil storage capacity. In contrast, lower elevation areas dominated
by grasslands and wetlands benefit from gentler slopes and deeper root zones, which
enhance infiltration and delay runoff responses (Caviedes-Voullième et al., 2021).

These findings support the view that vegetation functions as a spatially variable
regulator of runoff generation, contingent on topographic context and soil–plant–
atmosphere interactions. This regulatory effect is particularly sensitive to future
changes in vegetation cover and distribution under climate and land use change
scenarios. For instance, overgrazing may reduce root zone storage capacity, thereby
increasing runoff and erosion risks (Donovan and Monaghan, 2021). while shifts in
vegetation type (e.g., shrub encroachment or forest decline) could alter hydrological
partitioning along elevation gradients (Hsu et al., 2025; Zhou et al., 2023).

**6 Conclusions**
Hydrological modelling in high-latitude regions poses considerable challenges due to
the complexity of cryospheric processes and limited observational data. To address
these issues, this study proposes a stepwise modelling framework that incrementally
refines model structures by incorporating key hydrological processes and landscape



characters, thereby enhancing both the physical realism and predictive performance
of the model.

Our results underscore the limitations of the lumped models (FLEX$^L$ and FLEX$^L$-S) in
accurately representing runoff dynamics, particularly in regions with complex
topography and heterogeneous vegetation cover. Although the distributed model
FLEX$^D$ improved the simulation of runoff variability by incorporating spatially
distributed inputs, it still lacks full physical interpretability of its parameters. In contrast,
the landscape-based FLEX$^T$ model explicitly integrates snowpack, topography, and
vegetation characteristics, thereby enhancing the physical realism of parameterization
and offering a more mechanistic representation of hydrological processes. While
FLEX$^T$ achieved performance comparable to FLEX$^D$ in simulating catchment runoff
dynamics, this outcome may be attributed to the limited vegetation heterogeneity in the
study basins. Nonetheless, validation using SWE confirmed FLEX$^T$'s capability to
capture seasonal patterns, interannual variability, and key hydrological mechanisms in
cryospheric environments. These findings underscore the potential advantages of
FLEX$^T$, particularly in basins with greater ecological or topographic complexity.

Results from the FLEX$^T$ model indicate that snowmelt contributes 23.5%±1.3% and
14.7%±1.6% to streamflow in the Bogd Uliastai and Zavkhan Guulin river basins,
respectively. Temporally, snowmelt contributions peak in spring and remain minimal
during other seasons. Spatially, snowmelt contributions increase with elevation,
underscoring the critical role of topography in shaping the spatiotemporal dynamics of
runoff generation. In high elevation areas, the lagged snowmelt response leads to a
sustained and gradual release of runoff, whereas low-altitude areas respond more
rapidly to rainfall events. Moreover, hydrological modelling approaches based on
vegetation landscape classifications better capture spatial heterogeneity and
characterize the dominant hydrological mechanisms across different landscape units.
These findings offer valuable insights into hydrological response mechanisms in cold



alpine basins with limited observational data on the Mongolian Plateau. The stepwise
modeling framework developed in this study not only improves the simulation of runoff
dynamics in high-latitude regions but also enhances understanding of cryospheric
hydrological responses to global climate change. Importantly, this framework holds
both scientific and practical value, providing a foundation for more effective water
resource management, ecological conservation, and climate adaptation in cryospheric
and data-scarce regions.

**Code availability**

The code is available upon request to the contact author.

**Data availability**

All data presented in this manuscript are publicly available for download from:
Hydrometeorological data (precipitation, runoff, temperature) from the Information and
Research Institute of Meteorology, Hydrology, and Environment, available at
http://irimhe.namem.gov.mn. Arctic Snow Water Equivalent Grid Dataset from the
National Tibetan Plateau/Third Pole Environment Data Center, available at
https://cstr.cn/18406.11.Snow.tpdc.271556. Shuttle Radar Topography Mission Digital
Elevation Model (SRTM-DEM), available at http://srtm.csi.cgiar.org. Sentinel-2 10-
meter Land Use/Land Cover data from the ESRI Living Atlas, available at
https://livingatlas.arcgis.com/landcover/. NDVI data were obtained from the United
States Geological Survey (USGS) EarthExplorer platform. available at
https://earthexplorer.usgs.gov/.

**Author contributions**

LY and HG designed the study. BD provided the valuable fieldwork data. LY, YW, HG,
and ZD conducted the analyses. LY wrote the paper. All authors discussed the results
and the first draft and contributed to the final paper.



**Competing interests**

At least one of the (co-)authors is a member of the editorial board of Hydrology and Earth System Sciences.

**Acknowledgments**

This research is funded by National Key Research and Development Program of China (2024YFE0113200), the National Natural Science Foundation of China (grant no. 42471040). Zheng Duan would like to acknowledge the support from the Crafoord Foundation, Sweden (Grant No.20210552 and No.20240857). This work was performed as part of the IAHS HELPING Working Group on "Development & application of river basin simulators".

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
