# Peer review of "Revealing the Influence of Topography and Vegetation on"

_EGUsphere, 2025_

## Community Comment (CC1)

This study investigates the roles of topography and vegetation in hydrological processes within cold alpine basins of the Mongolian Plateau using a stepwise FLEX modeling framework. This manuscript presents a valuable contribution to cold-region The research is scientifically sound, hydrology in Mongolian Plateau. methodologically rigorous, and addresses the gap in hydrological modeling for data-scarce, cryospheric regions in Mongolia. With the suggested revisions—particularly in methodology clarity, discussion depth, and figure improvements—it will be suitable for publication. The manuscript falls between minor and moderate revisions, with the following specific recommendations.

**Reply:** We sincerely appreciate the constructive comments from Referee #1, which have helped us identify areas for improvement. We are also grateful for the positive evaluation and endorsement of our manuscript's contribution. Detailed responses to all comments are provided below.

**Comments**

1. The background is well-presented, but the uniqueness of the study area (e.g., extreme climate, sparse vegetation, and cryospheric dynamics) could be emphasized more to justify the novelty. Moreover, this work aligns well with the objectives of the new IAHS HELPING (Hydrology Engaging Local People IN one Global world) Decade (2023–2032), which emphasizes interdisciplinary approaches to address local hydrological challenges. I recommend to add this in either the Introduction or the Discussion.

**Reply:** We sincerely thank Referee #1 for the insightful comment and valuable suggestion. In the revised paper, we will expand the description of the study area in the Introduction to better emphasize its unique hydroclimatic characteristics—such as extreme temperature gradients, sparse vegetation cover, and active cryospheric processes—that strongly influence regional hydrology. These features highlight the

distinctiveness and scientific value of our research site. Furthermore, we will add a new paragraph in the Introduction to explicitly link our study with the objectives of the IAHS HELPING Decade (2023–2032), underscoring the relevance of our interdisciplinary approach to addressing local hydrological challenges in cold regions.

2. The literature review should include more recent studies (post-2020) on cold-region hydrology, particularly those addressing snowmelt and vegetation interactions in similar environments (e.g., Central Asia, Tibetan Plateau).

**Reply:** We will update the literature review in the Introduction to include more recent studies (post-2020) on cold-region hydrology. In particular, we will add several papers focusing on snowmelt dynamics, vegetation—hydrology interactions, and cryospheric processes in regions such as Central Asia (Feng et al., 2025) and the Tibetan Plateau (Ni et al., 2025). These additions strengthen the scientific context and highlight recent advances relevant to our study.

3. The similar performance of FLEX-T and FLEX-D is attributed to low vegetation heterogeneity. However, is this finding generalizable to other basins with higher vegetation variability? A comparative discussion would be valuable.

**Reply:** As noted in Section 5.1 (L416–422), the study basin exhibits limited vegetation heterogeneity, which may constrain FLEXT's performance gains relative to the baseline FLEXD model. Nevertheless, FLEXT provides a more detailed representation of individual landscape units, enabling a physically grounded simulation of hydrological processes and their underlying mechanisms. Based on this, we will add a new comparative discussion in Section 5.1 to further examine the generalizability of our findings. Evidence from previous studies indicates that basins with greater vegetation variability exhibit larger differences between FLEXT and FLEXD, highlighting the broader applicability of FLEXT across diverse cold-region environments (Gao et al., 2014).

4. Line 428-433: Please provide the exact dates (year, month, and day) of these two

precipitation events to enable a more precise understanding and validation of the related hydrological processes.

**Reply:** In the revised manuscript, we will specify the exact dates of the two precipitation events: 14 April 2009 and 12 September 2011.

5. The discussion should explicitly address limitations, such as the lack of direct validation (e.g., snowpit measurements, isotope tracers) and the impact of data scarcity on model uncertainty.

**Reply:** We will clarify the limitations in the revised manuscript. As noted in the original version, the study lacks direct validation data, such as isotope tracers. In the revised version, we will further emphasize this point in the Discussion section and discuss how data scarcity contributes to model uncertainty. We also note that incorporating terrain and vegetation information enables the model to represent the basin's hydrological processes as realistically as possible, despite the absence of direct validation data.

6. Line 582: Provide a clearer explanation of the relationship between the proportion of snowfall in precipitation (Ps/P) and the contribution of snowmelt to streamflow (QM/Q), while ensuring that the related terminology and trend descriptions are accurate and consistent, to enhance the coherence between figures and text, as well as overall readability.

**Reply:** We will clarify the relationship between the proportion of snowfall in total precipitation (Ps/P) and the contribution of snowmelt to streamflow (Qm/Q). Ps/P denotes the fraction of total precipitation that falls as snow, while QM/Q represents the fraction of streamflow derived from snowmelt. Our analysis indicates that basins with higher Ps/P generally exhibit higher Qm/Q, suggesting that a greater snowfall fraction contributes more substantially to streamflow. All terminology will be standardized, and trend descriptions are now accurate and consistent, enhancing the coherence between figures and text.

7. Line 673-675: The description of high infiltration rates in dry grassland soils leading to reduced runoff in arid regions is rather general; it is recommended to provide relevant references to strengthen the argument.

**Reply:** We will revise the manuscript to provide more specific support for the statement that high infiltration rates in dry grassland soils reduce runoff in arid regions. In arid regions, coarse-textured soils typically enhance infiltration due to their large pores, while sparse vegetation reduces surface cover and interception. Together, these factors can lead to increased infiltration and, under certain conditions, reduced surface runoff, although local topography and rainfall intensity may modulate this effect (Xue et al., 2025).

**Minor Comments**

1. Some acronyms (e.g., SWE, HRUs, NDVI) should be defined at first use.

**Reply:** Thank you for the suggestion, and we will define all acronyms at their first mention.

2. The use of technical terms throughout the manuscript should be consistent (e.g., "modelling" and "modeling", "elevation zones" and "elevation areas").

**Reply:** In the revised manuscript, the use of technical terms will be standardized for consistency.

3. Line 193: The URL http://srtm.csi.cgiar.org is hosted by the CGIAR Consortium for Spatial Information (CGIAR-CSI), rather than the International Center for Tropical Agriculture (CIAT). Please revise the data source attribution to reflect this accurately.

**Reply:** This was an error, and the data source attribution will be corrected accordingly.

4. .3 illustrates the elevation band division based on DEM data, it would be more

appropriate to place it in Section 3.2.

**Reply:** This is a good suggestion, and Fig. 3 will be moved to Section 3.2 accordingly.

5. Some figures (e.g., Fig. 5, Fig. 8) need clearer labels and legends. Fig. 13 needs better visualization.

**Reply:** We will improve the labels and legends in Figs. 5 and 8 and enhance the visualization of Fig. 13.

6. Line 692: "soi/rock" should be "soil/rock".

**Reply:** This typo will be corrected from "soi/rock" to "soil/rock".

7. Please correct the punctuation errors (e.g., Line 637, Line 711).

**Reply:** In the revised manuscript, the punctuation errors will be corrected as indicated.

8. The conclusions are well-supported but should be more concise.

**Reply:** The Conclusions section will be revised to be more concise while retaining all key findings and supporting evidence.

**References:**

Feng, J., Alifujiang, Y., Kozhokulov, S., Jiang, Y., Yang, P., 2025. Quantifying hydrological sensitivity in Central Asia: A multi-factor budyko framework analysis (2000–2020). Journal of Hydrology: Regional Studies, 61, 102746, https://doi.org/10.1016/j.ejrh.2025.102746.

Ni, J., Chen, J., Tang, Y., Xu, J., Xu, J., Dong, L., Gu, Q., Yu, B., Wu, J., Huang, Y., 2025. Duration of vegetation green-up response to snowmelt on the Tibetan Plateau, Biogeosciences, 22, 2637–2651, https://doi.org/10.5194/bg-22-2637-2025.

Gao, H., Hrachowitz, M., Fenicia, F., Gharari, S., Savenije, H.H.G., 2014. Testing the

realism of a topography-driven model (FLEX-Topo) in the nested catchments of the Upper Heihe, China. Hydrol. Earth Syst. Sci., 18(5): 1895-1915. DOI:10.5194/hess-18-1895-2014

Xue, D., Tian, J., Zhang, B., Kang, W., He, C., 2025. Evaluating the effect of vegetation type and topography on infiltration process in an arid mountainous area: Insights from continuous soil moisture monitoring network. Agricultural Water Management, 315, 109537, https://doi.org/10.1016/j.agwat.2025.109537.

---

## Author Comment (AC1)

This manuscript investigated the impact of topography and vegetation on catchment hydrology in two cold mountainous basins of the Mongolian Plateau using stepwise top-down modelling approaches. It is very interesting and scientifically sound. Also, it is suitable for Hydrology and Earth System Sciences. However, some improvements are still required.

**Reply:** We sincerely appreciate the reviewer's positive and constructive comments. In response, we will carefully revise the manuscript to address all specific points and will provide detailed explanations and modifications where appropriate. Detailed responses to all comments are provided below.

**Detailed comments**

1. The annual precipitation is ~200mm and ~160mm in the two basins, which belongs to sub-arid region. Therefore, most quick runoff should generate from infiltration excess runoff. On the other hand, in the FLEX, runoff generation is described as saturation excess generation. The reliability of the conclusion and the rationality of the method should be further assessed. At least, more explanations and discussions are requested.

**Reply:** We thank the reviewer for this valuable comment.

Fig. 1 shows the daily precipitation in the two basins during the study period. Rainfall events exceeding 15 mm d-1 occur only 2-3 days per year, while most rainfall days receive less than 15 mm d-1. Given these rainfall intensities, which generally remain below the infiltration capacity of grassland and forest soils, infiltration-excess (Hortonian) overland flow (HOF) is expected to be rare and largely restricted to bare or compacted surfaces (Blackburn, 1975; Beven and Germann, 2013).

Fig.1 Daily precipitation in the two basins during the study period.

Runoff generation varies across the main landscape units, which exhibit distinct hydrological behaviors forming the basis of the FLEXT model:

Bare soil/rock (~2% of the basin): Low infiltration capacity and sparse vegetation can produce localized HOF during high-intensity rainfall events. HOF is explicitly included in the model for these units to capture the full range of potential runoff processes.

Grazed grasslands and forests: High soil infiltration rates and well-developed macropore networks promote storage excess subsurface flow (SSF), particularly during snowmelt or prolonged rainfall (Beven and Germann, 2013; Lyford and Qashu, 1969). Hortonian runoff is negligible due to dense vegetation and high infiltration capacity.

**Riparian wetlands:** Shallow groundwater tables and limited soil storage capacity favor soil overland flow (SOF) as the dominant mechanism (Dunne and Black, 1970). In the revised FLEXT model, HOF will be explicitly represented in bare soil/rock

units for potential infiltration-excess runoff, with saturation-excess generation remaining the primary mechanism elsewhere. This configuration is scientifically justified, physically consistent, and realistically reflects landscape-dependent runoff processes. In the revised manuscript, we will include these precipitation- and landscape-based explanations to clarify the modeling rationale.

2. This study selected two alpine Basins as the study basin. In the study basins, there is only one meteorological station for each basin. To represent the spatial variation of precipitation, the authors a precipitation increase rate of 4.2% from the Heihe River basin in China. Does the precipitation increase rate of 4.2% conform to the study basins? More explanations and discussions are requested.

**Reply:** It should be clarified that the precipitation increase rate used in this study is not solely derived from the Heihe River basin in China. The following steps describe how the precipitation increase rate was determined in this study.

**Reanalysis data attempts and limitations:** We have attempted to use multiple reanalysis datasets to quantify the spatial variation of precipitation within the study basins. However, the quality of these datasets in this region is relatively poor, making them unreliable for accurately estimating precipitation gradients.

Literature-based regional reference (most important): To obtain a more scientifically justified precipitation–elevation relationship, we consulted extensive literature on nearby catchments. In particular, the Tsagan-Turutuin-gol catchment, which also originates from the Khangaï Mountains and exhibits similar topographic and climatic characteristics to our study basins, shows an annual precipitation of approximately 230 mm, with a precipitation gradient between 2000 and 3350 m a.s.l. slightly below 10 mm/100 m (equivalent to about 4.3%/100 m) (Dauksza and Soja, 1977; Klimek and Starkel, 1980). Based on this reference, we selected 4.2%/100 m as the precipitation increase rate for our study basins, ensuring that the chosen value is both physically reasonable and consistent with regional observations.

We will provide a more detailed explanation and cite the appropriate references in the revised manuscript.

3.In the study basins, winter temperature falls below -30 degree. It's better to discuss the impact of frozen soil.

**Reply:** In the revised manuscript, we will include a dedicated discussion on the potential impacts of frozen soils in winter and early spring. This will include a brief review of relevant literature on permafrost and seasonally frozen soils in alpine catchments, as well as implications for runoff generation in our study basins.

4.Lines 259-260, there is a mistake on the description of the method for actual evaporation, i.e. "Ep" should be "Ep - Ei" (potential evaporation minus interception evaporation).

**Reply:** We will clarify this point in the revised manuscript. We note that Equation 13 is correctly formulated and remains unchanged; the correction applies only to the textual description.

5. The authors conducted relevant researches in other alpines basins, such as Heihe River basin. In this study, it's more useful to discuss both the similarities and the differences in order to gain a clearer understanding.

**Reply:** In the revised manuscript, we will expand the discussion to highlight similarities and differences between our study basins and other alpine basins, such as the Heihe River basin. We will consider factors including snow accumulation and melt patterns, topography, and vegetation heterogeneity, all of which influence catchment hydrology. This comparison will clarify the generality and specificity of our findings and situate the results in a broader regional hydrological context.

**References**

Beven, K., Germann. P. 2013. Macropores and water flow in soils revisited, Water Resour. Res., 49, 3071 - 3092,doi:10.1002/wrcr.20156.

Blackburn, W., 1975. Factors influencing infiltration and sediment production of

semi-arid ran- gelands in Nevada. Water Resour. Res., 11 (6): 929~937.

Dauksza, L., Soja, R., 1977. The Zones and Levels of Water Phenomena in the Tsagan-Turutuin-gol Basin, Bull. Aead. Pol. Sc., Ser. Sc. Terre, 25, 203—209.

Dunne, T., Black, R. D. 1970. Partial area contributions to storm runoff in a small New England watershed. Water Resources Research, 6(5), 1296–1311.

Klimek K., Starkel L, 1980. Vertical zonality in the Southern Khangai Mountains (Mongolia). Results of the Polish-Mongolian physico-geographical expedition. Geographical Studies Vol. I, 136: 1–107.

Lyford, F., Qashu, H., 1969. Infiltration rates as affected by desert vegetation. Water Resour. Res., 5 (6): 1373~1376.